# High-Fidelity Synthetic Transmission Electron Microscopy Image Generation Using Diffusion Probabilistic Models for Data-Limited Semiconductor Metrology

## Abstract

Advanced semiconductor nodes have significantly increased the demand for Transmission Electron Microscopy (TEM) characterization, posing unprecedented challenges for metrology extraction and defect inspection due to device complexity and shrinking critical-dimension (CD). However, the destructive nature of TEM sample preparation, combined with time-intensive imaging, high acquisition costs, and reproducibility issues, severely limits the availability of diverse datasets required for conventional experimental analysis and, consequently, for training machine learning (ML) models. As a result, artificial intelligence-based synthetic data generation and augmentation have become essential in semiconductor TEM research. Existing generative approaches often fail to capture the complex noise characteristics, surface features, and stochastic variations present in real TEM images, which are critical for accurate semiconductor metrology evaluation. In this research, we present a novel generative framework utilizing Denoising Diffusion Probabilistic Models (DDPMs) specifically designed for synthetic TEM image generation under extreme data scarcity. Our approach employs a progressive patch-based training strategy that scales from low-resolution patches to full-resolution images, enabling from-scratch model training with datasets containing as few as 15 images. We integrate a custom adaptation of the TrivialAugment (TA) algorithm, incorporate domain transfer for cross-process compatibility, and apply super-resolution-enabling inpainting techniques alongside classifier guidance. This framework culminates in full-image training, generating coherent high-resolution TEM images that preserve global structural and spatial relationships essential for analyzing large-scale device structures, in compliance with real FAB semiconductor metrology requirements. Our synthetic images are visually indistinguishable from real TEM data, with a high structural similarity index (MS-SSIM $> 0.94$) confirming their high-fidelity reproduction, as validated by domain experts. The generated synthetic datasets aim to facilitate robust training of downstream ML models for defect detection, grain and phase boundary segmentation, and metrology applications, addressing critical data availability bottlenecks in advanced semiconductor manufacturing while preserving the statistical and physical properties of scarce real TEM imaging data.

## 1 Introduction

The relentless advancement of semiconductor technology has driven feature sizes to unprecedented scales, with advanced node technologies now reaching dimensions of 2nm and beyond. This continuous miniaturization has been enabled by advanced lithography, particularly Extreme-Ultra-Violet-Lithography (EUVL) and the emerging high-NA EUVL, which are essential for printing the increasingly smaller features required by modern integrated circuits. However, as feature dimensions continue to shrink, the challenges associated with process control, defect inspection, and metrology have escalated substantially in complexity and are expected to intensify further (Lorusso et al., 2022; Chen et al., 2024).

Transmission Electron Microscopy serves as an indispensable metrology tool in advanced semiconductor node design and manufacturing, delivering atomic-scale resolution imaging for visualizing and analyzing nanometer-scale structures and defects. The acquisition of high-quality TEM images is critical for various applications including defect classification, thickness measurements, and process optimization, but the generation of comprehensive TEM datasets faces practical limitations. As an inherently characterization technique, TEM requires extensive sample preparation that renders devices permanently unusable for production, limiting the availability of adequate and diverse samples for comprehensive analysis. The time-intensive nature of TEM imaging, combined with the expertise required for sample preparation and data acquisition, creates substantial bottlenecks in processes development.

Scarcity of high-quality TEM datasets presents a fundamental challenge to the development of both conventional and ML-based analysis tools. While domain experts may question the usefulness of synthetic images, recent evidence demonstrates their effectiveness in improving performance for downstream tasks (Govind et al., 2024). Recent developments in generative artificial intelligence, particularly DDPMs, have shown great success in generating high-fidelity synthetic images across various domains (Ho et al., 2020; Nichol & Dhariwal, 2021; Dhariwal & Nichol, 2021). Their application to semiconductor imaging has been explored for scanning electron microscope (SEM) based defect inspection (De Ridder et al., 2023; Dey et al., 2024), demonstrating their potential. Unlike SEM imaging, which is primarily surface-sensitive and non-destructive, TEM imaging enables transmission imaging through ultra-thin samples of $<100$ nm with sub-angstrom resolution, making it suited for analyzing internal device structures and atomic-scale features. Extending generative techniques to TEM imaging faces unique challenges, including the severe scarcity of well-annotated datasets and the need to generate high-resolution images that faithfully preserve critical semiconductor metrology-aware structural features. While data augmentation techniques have been shown to substantially enhance machine learning model training in electron microscopy domains (Shaga Devan et al., 2021; Ede, 2021; Chen & Barnard, 2024; Kazimi et al., 2024), the application of diffusion models for robust TEM image synthesis remains largely unexplored and represents a highly promising avenue for further research.

In this research, we present a novel approach for generating synthetic TEM images using custom-trained diffusion models specifically designed to operate effectively with limited real wafer TEM datasets. Our methodology addresses the critical data scarcity challenge in semiconductor TEM imaging by training generative models from scratch, enabling the generation of domain-specific synthetic images that preserve the characteristics of real TEM data. Our approach employs a progressive training strategy, that builds from small image patches to full-scale, high-resolution images while ensuring accurate reproduction of both fine-grained textures and large-scale structural features. The generated images remain grounded in the original data's distribution, preserving statistical and structural characteristics of real TEM data while providing the necessary diversity for effective ML model training to achieve robustness and generalization. While previous research efforts and trained models predominantly focus on downstream tasks and post-acquisition analysis, with the aid of existing methodologies, our generative framework can subsequently accelerate acquisition processes and enable automated workflows (Spurgeon et al., 2021; Botifoll et al., 2022; Kalinin et al., 2022a;b; Meirovitch et al., 2023; Kalinin et al., 2023).

The main contributions of this research are: **(i)** the development of a patch-based progressive training methodology for denoising diffusion probabilistic models (DDPMs) that enables the generation of high-fidelity, high-resolution synthetic TEM images from extremely limited datasets, including scenarios with as few as 15 training images. This approach produces synthetic images that are largely indistinguishable from real wafer TEM data and meet semiconductor metrology specifications; **(ii)** adoption of the TA algorithm (Müller & Hutter, 2021) to TEM imaging and the integration of semiconductor imaging-aware augmentation methods (Chen et al., 2024); **(iii)** the demonstration of domain transfer capabilities, facilitating the generation of synthetic TEM images that preserve characteristics from varying imaging conditions or process parameters, thereby enhancing model adaptability across different semiconductor manufacturing contexts; **(iv)** the integration of guidance mechanisms within the generative process to enable controllable synthesis, allowing the generation of synthetic images with specific desired structural properties relevant to semiconductor metrology; **(v)** the adaptation and application of state-of-the-art inpainting capability (Lugmayr et al., 2022), within the diffusion modeling framework, enabling seamless image extension and resolution enhancement of synthetic TEM images.

## 2 RELATED WORK

### 2.1 SIMULATION-BASED

TEM image synthesis primarily relies on supervised methods like physics-based simulations modeling electron-specimen interactions. Several packages provide frameworks, including abTEM (Madsen & Susi, 2021), Prismatic (Rangel DaCosta et al., 2021), and conventional multislice algorithms (Kirkland, 2020)[p. 180]. They allow precise control over imaging parameters and deliver physically accurate representations of electron scattering processes. However, these methods face limitations when applied to large-scale device structures in semiconductors, as they require extensive prior knowledge of specimen composition, imaging conditions, and microscope parameters, which may not always be available. While simulation frameworks such as Construction Zone (Rangel DaCosta et al., 2024) have demonstrated success in generating synthetic datasets for nanoparticle segmentation, conventional simulations remain constrained by their reliance on predefined physical models. They may not capture the full complexity of real experimental conditions and become increasingly computationally demanding as the simulated specimen size grows. Consequently, they are not well-suited for generating large-scale datasets intended for training ML models (Botifoll et al., 2022).

### 2.2 RULE-BASED AND CONCEPT-ORIENTED

Rule-based semi-supervised approaches represent another category of synthetic TEM image generation, relying on predefined structural concepts and procedures. Govind et al. (2024) proposed a parametric model for generating synthetic training data for dislocation segmentation, demonstrating that synthetic images can yield superior performance compared to models trained solely on real data. Saleh et al. (2025) introduced a concept-oriented approach that integrates edge detection with diffusion models for denoising and connecting broken segments. They use diffusion models primarily for post-processing and noise reduction rather than direct image generation. While rule-based approaches offer precise control over structural features and enable efficient large-scale dataset generation, they remain constrained by the predefined concepts and rules encoded in their algorithms and thereby, like simulations, may fail to capture the full complexity and stochastic variability observed under real experimental conditions.

### 2.3 GENERATIVE ADVERSARIAL NETWORKS

Recently, Generative Adversarial Networks (GANs) have been introduced as an unsupervised deep-learning approach for synthetic Electron Microscopy image generation. Khan et al. (2023) demonstrated the use of cycle-consistent GANs to generate realistic scanning transmission electron microscopy images (STEM), while Eliasson et al. (2025) employed CycleGANs to bridge experimental and simulated images for catalyst nanoparticle analysis with high-angle annular dark-field STEM. In the biological domain, Shaga Devan et al. (2021) showed that GANs could effectively augment transmission electron microscopy datasets, improving automated detection performance by generating synthetic labeled images. While GAN-based approaches are capable of generating visually realistic images, they are susceptible to mode collapse and training instabilities, which can limit the diversity and quality of generated samples. Additionally, GANs often face challenges preserving the fine-grained structural details that are critical for quantitative analysis (Dhariwal & Nichol, 2021).

### 2.4 DIFFUSION MODELS

The emergence of DDPMs represents a paradigm shift in generative modeling for unsupervised image synthesis. Initially introduced by Ho et al. (2020) and popularized by implementations such as DALL-E (Ramesh et al., 2021) and Stable Diffusion (Rombach et al., 2022), DDPMs formulate image generation as a gradual denoising process, in which models are trained to reverse a diffusion procedure that progressively corrupts training images with Gaussian noise (Sec. A.1). Improvements by Nichol & Dhariwal (2021) enhanced their quality, while the introduction of Denoising Diffusion Implicit Models (DDIM) (Song et al., 2022) addressed the sampling efficiency during inference. Furthermore, Dhariwal & Nichol (2021) demonstrated that diffusion models achieve superior image quality compared to GANs on standard benchmarks. Diffusion models offer several advantages, including greater training stability, improved mode coverage, preservation of fine-grained structural

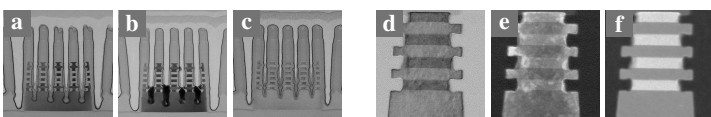

Figure 1: (a–c) Structures from DEVICE-TEM; (d–f) BF, ADF, HAADF modes from NANO-TEM.

details, and enhanced generative diversity. Their iterative denoising mechanism also captures noise characteristics and imaging artifacts inherent in real data, making them particularly well-suited for generating synthetic datasets that retain the statistical properties of experimental measurements obtained from actual measurements. To the best of our knowledge, the work presented in this paper is among the first to investigate their application in TEM imaging.

## 3 DATA

We use high-resolution TEM images from semiconductor manufacturing research, acquired using actual FAB tools and stored as $3 \times 8$ bit grayscale JPEG files. They capture nanometer-scale device structures and potential defects representative of advanced node manufacturing processes.

**DEVICE-TEM Dataset:** This dataset comprises 15 TEM images, of which 9 with a resolution of $(4096 \times 3874)$ pixels, 6 at $(2048 \times 1936)$ pixels, and 3 at $(2048 \times 735)$ pixels. These lower-magnification images $(200,000\times$ to $300,000\times)$ capture multiple structural features such as fins, cavities, and pillars, providing broader context for process-related geometries including cavity depth, sidewall profiles, and hard-mask integrity, critical for across-feature metrology and process variation studies. Three different imaging modes are featured: bright field (BF), annular dark-field (ADF) and high-angle annular dark field (HAADF), covering three distinct structures as shown in Fig. 1 (a-c).

The higher-magnification dataset (NANO-TEM) is depicted in Fig. 1 (d-f) and detailed in Sec. A.3.

## 4 METHODOLOGY

### 4.1 PATCH-BASED PROGRESSIVE TRAINING

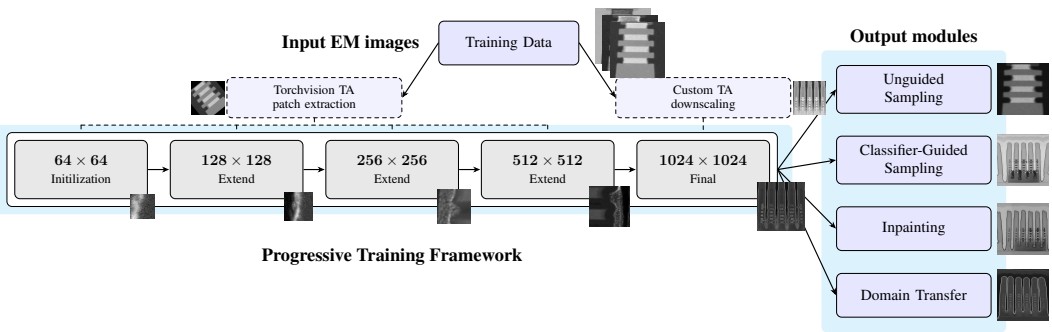

Figure 2: Framework of the proposed patch-based training process.

We implement patch-based progressive training as depicted Fig. 2, systematically increasing both spatial resolution and dataset size, transitioning from fine-grained textures at small scales to global structural features at larger scales. The training progresses through multiple resolution stages of $(64 \times 64)$, $(128 \times 128)$, $(256 \times 256)$, and $(512 \times 512)$ pixel patches, before finally moving to full perspective images downscaled to $(1024 \times 1024)$ resolution. From each training image, patches are extracted to maximize data utilization while preserving spatial coherence, allowing the integration of details that would otherwise be lost if training began directly on downscaled, full-perspective images. The extraction strategy ensures comprehensive coverage of image content while providing sufficient training samples. Edge regions were upscaled using Lanczos filtering (Duchon, 1979)

to preserve texture. We trained a separate model for each dataset. Sec. A.10 provides an ablation study for the proposed patch-based incremental training approach, compared against baseline DDPM model trained for target resolution ($128 \times 128$), respectively.

## 4.2 DATA AUGMENTATION

We implement a custom adaption of the TA algorithm (Müller & Hutter, 2021), tailored for semiconductor TEM imaging. It is detailed in Sec. A.4. TA offers a simple yet effective augmentation strategy by randomly selecting both the augmentation operation and its magnitude for each training sample, thereby avoiding the overhead of hyperparameter tuning. Our implementation incorporates grayscale-specific modifications and rejection criteria for extreme brightness values and hash-based duplicate filtering to ensure structural integrity and maintain control over the distribution of optical characteristics such as brightness and contrast. For the final training stage, we transitioned from the standard TorchVision TA implementation to a customized approach that incorporates augmentation strategies suggested in Chen et al. (2024), built on Albumentations and OpenCV. This prevents destructive transformations that could compromise the structural integrity of full-resolution images. Tab. 1 summarizes the dataset sizes after separately applying $20\times$ augmentation to each stage.

Table 1: Dataset sizes after $20\times$ augmentation

| Resolution | DEVICE-TEM | NANO-TEM |
|---|---|---|
| $64 \times 64$ | 877,307 | 11,590,656 |
| $128 \times 128$ | 219,327 | 2,897,664 |
| $256 \times 256$ | 54,832 | 724,416 |
| $512 \times 512$ | 13,708 | 181,104 |
| $1024 \times 1024$ | 315 | 11,319 |

## 4.3 DIFFUSION MODEL ARCHITECTURE AND TRAINING

### 4.3.1 MODEL ARCHITECTURE

We employ a DDPM based on work from Nichol & Dhariwal (2021) in a U-Net configuration (He et al., 2016), implemented in PyTorch. The model is constructed with increasing depth to enable seamless weight transfer across resolution stages without requiring architectural modifications. The U-Net features a symmetric encoder-decoder architecture with skip connections, facilitating the preservation of fine-grained details. Attention mechanisms are incorporated at multiple resolution levels to capture both local texture patterns and global structural information.

### 4.3.2 HYPERPARAMETERS AND TRAINING CONFIGURATION

Sec. A.2 details the architecture used for each training stage. The channel multipliers and residual block configurations are chosen to accommodate the progressive resolution increase. Patch-based training is performed for 10 epochs per patch-size. Our experiments indicated that the final stage required significantly more epochs, reflecting the increased complexity of high-resolution features. Computational requirements for training and inference are detailed in Sec. A.5.

### 4.3.3 PROGRESSIVE TRAINING AND WEIGHT TRANSFER

For each resolution stage beyond the initial ($64 \times 64$), we freeze encoder layers for half of the training iterations, allowing the model to adapt decoder layers to the new (higher) resolution while preserving low-level features learned in earlier stages, improving knowledge transfer and training stability. We initialized the higher-resolution models with the complete set of weights from the preceding stage, ensuring architectural and parameter compatibility across all resolution levels. In Sec. A.10 we report an ablation study evaluating layer-freezing and weight transfer ($64 \times 64$ to $128 \times 128$), compared to baseline DDPM training at ($128 \times 128$) resolution without any freezing strategy. Our experiments show that the baseline DDPM without freezing consistently performs worst, while patch-based training without freezing yields the best overall results, strongly supporting the importance of the patch-based training strategy.

## 4.4 TRAINING OBJECTIVE AND LOSS FUNCTION

Following the standard DDPM formulation (Ho et al., 2020), our models are trained to predict the noise component $\epsilon$ added at each timestep $t$ of the diffusion process. The simplified training objective minimizes the mean squared error (MSE) between predicted and actual noise:

$$\mathcal{L} = \mathbb{E}_{t,x_0,\epsilon \sim \mathcal{N}(0,I)} \left[ \|\epsilon - \epsilon_\theta(x_t, t)\|^2 \right] \tag{1}$$

Here, $x_0$ represents the original TEM image, $\epsilon_\theta$ is the neural network parameterized by $\theta$, and $x_t$ is the noisy image at timestep $t$, see Sec. A.1. While the training objective remains consistent across all stages, our progressive training strategy facilitates hierarchical feature learning by progressively increasing data complexity and model scaling. At the ($64 \times 64$) resolution stage, training emphasizes image characteristics such as contrast, noise patterns, and basic textures. As the resolution increases, the model builds on these representations while incorporating larger-scale structural information, including boundaries and spatial relationships. Weight transfer and layer freezing ensure that fine-grained texture representations learned at lower resolutions are preserved and further refined.

## 4.5 STRUCTURAL SIMILARITY METRICS AND METHODOLOGY

**Metrics** To evaluate the structural similarity of original and synthetic images, we adopt domain-specific metrics proposed by Treder et al. (2022) and implemented by Detlefsen et al. (2022). As our augmentation algorithm 1 includes operations on brightness and contrast (Chen et al., 2024), we also employ the **Multi-Scale Structural Similarity Index Measure (MS-SSIM)** (Wang et al., 2003), improving robustness by computing the SSIM across multiple scales. We measure perceptual similarity by employing the **Learned Perceptual Image Patch Similarity (LPIPS)** metric proposed by Zhang et al. (2018) using the default AlexNet-backbone. The metrics are detailed in Sec. A.6.

**Methodology** We compute the similarity metrics on a class-wise basis using PSNR-sorted subsets of the synthetic datasets and report best results, which is explained by the unconditional architecture of our models. Since they were trained to generate diverse samples across the entire data distribution, evaluating against the entire synthetic dataset would inevitably include cross-class samples. Without explicit guidance, the generative process naturally shifts sample properties of generated samples along the distribution of the original dataset, occasionally merging class characteristics. A comprehensive evaluation under these conditions would therefore yield uninformative or misleading results. By selecting PSNR-sorted subsets, we focus on the highest-quality synthetic images that most closely align with real images in each class, providing a more meaningful assessment of the model's ability to generate structurally consistent outputs. We chose PSNR for this step because of its small computational overhead and strong correlation with SSIM and thus MS-SSIM (Horé & Ziou, 2010). Using the DEVICE-TEM model, 3,976 images were created for analysis. With the NANO-TEM model, 6,656 samples were generated. 250 DDIM timesteps and an $\eta$-value of 0.5, allowing for controlled stochasticity in the sampling process (Song et al., 2022), were used. Note that according to Sec. 5.2, an earlier stop might have given slightly better results.

**Baseline** For ablation and baseline comparisons, we employ MS-SSIM VAE (Snell et al., 2017) and DCGAN (Radford et al., 2016) architectures, trained from scratch using our augmentation and sampling strategies. We provide a comprehensive quantitative and qualitative assessment of similarity based on: (1) Pixel-level: PSNR, SSIM, MS-SSIM (Snell et al., 2017; Treder et al., 2022) (2) Feature/perceptual level: LPIPS (Zhang et al., 2018) (3) Task-specific: Noise distribution matching.

## 5 RESULTS

### 5.1 STRUCTURAL SIMILARITY (DEVICE-TEM)

Tab. 2 and Fig. 5 (a) summarize our similarity analysis. We report a mean MS-SSIM of 0.9435, indicating excellent similarity. As detailed in Sec. 4.5 and A.6, the MS-SSIM is particularly relevant for our application domain. The SSIM confirms strong performance with a mean value of 0.8416. PSNR results in a mean of 27.33 dB, considered acceptable. We attribute this to four factors: **(i)** the reverse diffusion process removes noise during generation, which smooths pixel-level details but

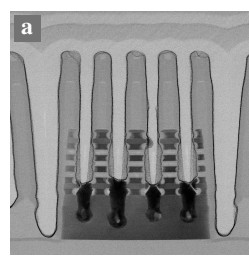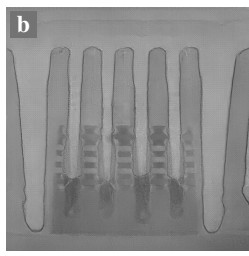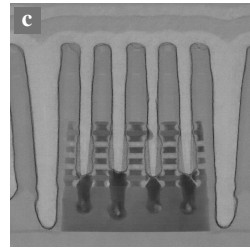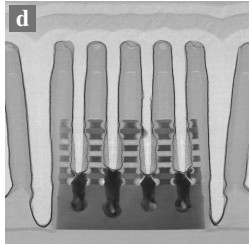

Figure 3: Original (a) DCGAN (b), MS-SSIM VAE (c) and proposed (d) generative approach.

Table 2: Structural similarity metrics for synthetic DEVICE-TEM PSNR-sorted subset ($n = 135$)

| Metric | DDPM (Ours) | MS-SSIM VAE | DCGAN |
|---|---|---|---|
| MS-SSIM | $0.9435 \pm 0.0201$ | $0.8951 \pm 0.0460$ | $0.3644 \pm 0.0040$ |
| SSIM | $0.8416 \pm 0.0455$ | $0.7013 \pm 0.1253$ | $0.2744 \pm 0.0069$ |
| PSNR (dB) | $27.330 \pm 0.0187$ | $25.080 \pm 0.0143$ | $17.910 \pm 0.0011$ |
| MSE | $0.0020 \pm 0.0007$ | $0.0032 \pm 0.0009$ | $0.0162 \pm 0.0004$ |
| LPIPS | $0.1466 \pm 0.0306$ | $0.3767 \pm 0.1330$ | $06213. \pm 0.0327$ |

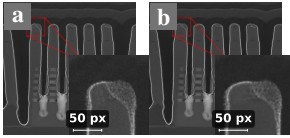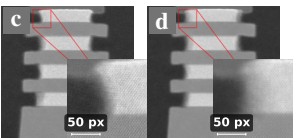

Figure 4: Original and synthetic images: DEVICE-TEM (a,b) and NANO-TEM (c,d).

maintains overall structural patterns; **(ii)** the JPEG compression applied to our training data introduces artifacts that influence pixel-wise comparisons; **(iii)** the stochastic sampling process prioritizes realistic texture generation over exact pixel reproduction; and **(iv)** the PSNR is very sensitive to variations in Gaussian noise (Horé & Ziou, 2010) which DDPMs inherently produce. The MSE remains low at 0.002, while an LPIPS score 0.1466 confirms high perceptual fidelity. Our extensive ablation study demonstrates that the proposed approach outperforms all baselines and well-established prior methods across every evaluated aspect, as depicted in Tab. 2 and Fig. 3. Although the best synthetic samples from the subset closely resemble the original images, they maintain subtle variations rather than being pixel-wise replicas, see Fig. 4. More synthetic samples are shown in Fig. 11. Overall, our analysis demonstrates a strong correspondence with real TEM Device images. Metrics for the NANO-TEM dataset are reported in Sec. A.7.

## 5.2 STRUCTURAL SIMILARITY AND INFERENCE TIME (DEVICE-TEM)

Fig. 5 (b) demonstrates the trade-off between image quality and inference time on a NVIDIA L40s GPU when varying the number of DDIM sampling steps. Similarity metrics were evaluated on the DEVICE-TEM dataset for 100 high-quality samples after each sampling timestep. They were selected based on MS-SSIM optimization across multiple random sampling seeds (different random initializations of the diffusion process). Quality improves rapidly within the first 50 steps and reaches near-maximum values well before the final step, indicating diminishing returns for additional steps.

## 5.3 PERCEPTUAL SIMILARITY (DEVICE-TEM)

15 original and 15 synthetic ($1024 \times 1024$) DEVICE-TEM images were submitted to five domain experts in an unordered manner for an uncontrolled blind evaluation. As the structural similarity results in Tab. 2 indicate, none of them were able to distinguish between real and synthetic images.

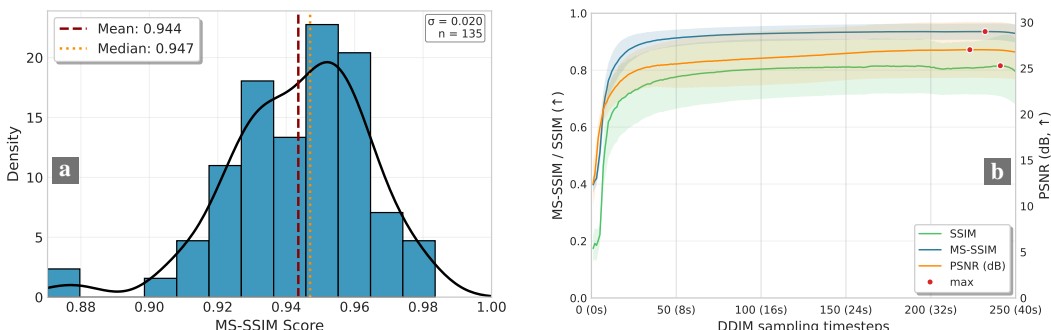

Figure 5: (a) DEVICE-TEM MS-SSIM; (b) similarity (mean $\pm$ std) vs. timesteps ($n = 100$).

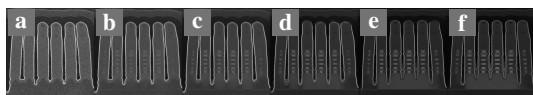

Figure 6: (a) Original; (b-f) domain transfer outputs after timesteps $t \in \{50, 100, 200, 300, 400\}$.

### 5.4 NOISE CHARACTERISTICS (DEVICE-TEM)

We computed three metrics, detailed in Sec. A.8, across the subset detailed in the previous sub-section. Tab. 3 summarizes our results. The synthetic images exhibit a significantly lower noise standard deviation ($0.008 \pm 0.003$ vs. $0.021\pm, 0.011$, $p < 0.001$). The small p-value indicates that the observed difference is statistically significant under the null hypothesis of no difference. However, the Signal-to-Noise Ratio (SNR) shows no significant difference ($24.0 \pm 2.9$ dB vs. $24.5 \pm 2.6$ dB, $p = 0.494$), indicating successful reproduction of the noise patterns. The ratio of high-frequency noise (HFNR), reflecting the spectral distribution of noise components, is slightly but significantly elevated in synthetic images ($1.190 \pm 0.007$ vs $1.141 \pm 0.032$, $p < 0.001$). This indicates that the noise in synthetic images differs subtly in its frequency characteristics, potentially due to the de-noising and smoothing effects of the reverse diffusion process. Our results demonstrate that this approach effectively preserves key domain-specific noise characteristics while generating images modestly denoised compared to the input. Such characteristics may be beneficial for downstream applications such as grain or layer boundary segmentation and defect classification, where reduced noise may enhance performance. Metrics for the NANO-TEM dataset are reported in Sec. A.9.

Table 3: Noise metrics for synthetic DEVICE-TEM PSNR-sorted subset ($n = 135$)

| Metric | Original | DDPM (Ours) | MS-SSIM VAE | DCGAN |
|--------|----------|-------------|-------------|-------|
| Noise std | $0.021 \pm 0.011$ | $\underline{0.008 \pm 0.003}$ ($p < 0.001$) | $0.009 \pm 0.003$ | $0.025 \pm 0.001$ |
| SNR (dB) | $24.0 \pm 2.9$ | $\underline{24.5 \pm 2.6}$ ($p = 0.494$) | $31.96 \pm 1.6$ | $26.15 \pm 0.1$ |
| HFNR | $1.141 \pm 0.032$ | $1.190 \pm 0.007$ ($p < 0.001$) | $1.214 \pm 0.003$ | $\underline{1.096 \pm 0.005}$ |

### 5.5 DOMAIN TRANSFER

We incorporate domain transfer capabilities into our framework, adapted from Smith et al. (2022), which enable the transformation of images across different imaging conditions or process parameters, while preserving high-level structural features. We apply the forward diffusion process to an input image $x_0$ for $T' \leq T$ timesteps, followed by reverse denoising using our trained DDPM. Specifically, given an input image $x_0$ outside the training set, we first apply noise according to:

$$x_{T'} = \sqrt{\bar{\alpha}_{T'}}x_0 + \sqrt{1 - \bar{\alpha}_{T'}}\epsilon \qquad (2)$$

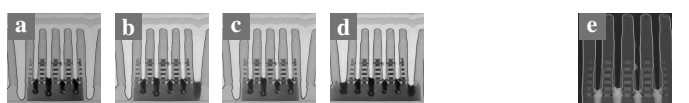

Figure 7: (a) Original; (b–d) guidance-scale ($s = 3, 5, 12$); (e) original, (f) mask, (g) inpainted.

where $\epsilon \sim \mathcal{N}(0, \mathbf{I})$ and $\bar{\alpha}_{T'} = \prod_{t=1}^{T'} \alpha_t$. The noisy image $x_{T'}$ is then processed through the reverse diffusion steps to generate an output adapted to the characteristics of the target domain, while maintaining structural properties of the input. The noise level parameter $T'$ governs the trade-off between domain adaptation and structural preservation. Higher values of $T'$ generate images closely aligned with the training distribution but with reduced retention of the original structure, whereas lower $T'$ values retain more structural details while still transferring domain-specific properties. As Fig. 6 demonstrates, this approach facilitates the transfer of structural features from a source to a target domain. It enables inter-domain transfer by adapting features from unseen input images (those not included in the training dataset and originating from different domains), to the target domain, aligning them with the characteristics of the training data. Furthermore, it supports the transfer of structural attributes across datasets derived from distinct training domains, for clear practical reasons. For semiconductor characterization, this addresses cross-tool and cross-process compatibility by generating images that incorporate characteristics from different imaging conditions, accelerating voltages, or specimen preparation protocols. This approach aims to reduce the need for extensive and potentially destructive TEM data acquisition under specific configurations, thereby accelerating analysis workflows and enhancing flexibility across different equipment setups. Validation of this claim requires comprehensive experimental studies beyond the scope of the present work.

## 5.6 CLASSIFIER GUIDANCE

We enable the controllable generation of specific classes through classifier guidance adapted from Dhariwal & Nichol (2021). This approach employs an external classifier trained on noisy images at multiple timesteps to guide the diffusion sampling process toward desired target classes. It modifies DDPM sampling by incorporating classifier gradients as follows:

$$x_{t-1} \sim \mathcal{N}\Big(\mu_\theta(x_t, t) + s\,\Sigma_t\,\nabla_{x_t} \log p_\phi(y \mid x_t, t),\ \Sigma_t\Big) \tag{3}$$

Here, $s$ describes the guidance scale, $p_\phi(y|x_t, t)$ represents the classifier's predicted probability for target class $y$, and $\nabla_{x_t} \log p_\phi(y|x_t, t)$ provides the guidance gradient. We trained a noise-aware classifier conditioned on timesteps, using the augmented DEVICE-TEM dataset. As illustrated in Fig. 7 (a-d), classifier guidance effectively steers the generation process toward specific classes, significantly enhancing class-conditional sample quality. Increasing the guidance scale improves class fidelity but tends to reduce the diversity of generated structures and textures. However, external classifiers present several limitations: (i) they require separate training and maintenance of classification models, (ii) the fidelity of guidance depends directly on classifier robustness and accuracy, and (iii) the two-stage training process increases computational complexity and resource demands. The effect of classifier guidance on sample distribution is visualized in Sec. A.11.

## 5.7 INPAINTING

Inpainting enables the reconstruction of masked or missing regions within an image while ensuring consistency with surrounding content. In the context of TEM applications, this capability facilitates image extension, super-resolution, and seamless integration of images acquired under varying conditions into unified datasets. We adapt the RePaint methodology (Lugmayr et al., 2022), leveraging the denoising process of DDPMs for conditional generation. In the reverse diffusion process, we combine information from known regions of the original image with predictions for unknown regions at each denoising step $t$:

$$x_{t-1}^{\text{known}} \sim \mathcal{N}(\sqrt{\bar{\alpha}_{t-1}} x_0, (1 - \bar{\alpha}_{t-1})\mathbf{I}) \qquad x_{t-1}^{\text{unknown}} \sim \mathcal{N}(\mu_\theta(x_t, t), \Sigma_\theta(x_t, t)) \tag{4}$$

$$x_{t-1} = m \odot x_{t-1}^{\text{known}} + (1 - m) \odot x_{t-1}^{\text{unknown}} \tag{5}$$

Here, $m$ represents the binary mask indicating known regions, $x_{t-1}^{\text{known}}$ is sampled using corresponding known pixels from original image $x_0$, while $x_{t-1}^{\text{unknown}}$ is generated by the diffusion model for unknown, masked regions. The mask ensures that known regions remain untouched while unknown regions are synthesized in a manner consistent with the learned imaging characteristics. This method enables several TEM-specific applications: **(i)** extension of smaller Field-of-View (FoV) images to facilitate broader structural analysis, **(ii)** super-resolution reconstruction as detailed by Lugmayr et al. (2022), and **(iii)** integration of images acquired at varying magnifications or imaging conditions. Fig. 7 (e-g) demonstrates the coherent reconstruction of a masked region. Here, 10 resampling iterations with a jump length of 10 steps, applied every 10 diffusion steps, produced the most coherent inpainting results. After 181 minutes of sampling on a NVIDIA L40S GPU, the reconstruction is perceptually indistinguishable from the original and achieves high fidelity, as indicated by the following metrics: MS-SSIM: 0.9901, SSIM: 0.9487, PSNR (dB): 37.0895, LPIPS: 0.0105.

## 6 LIMITATIONS AND FUTURE WORK

While we demonstrate successful synthetic TEM image generation, several limitations must be considered. With a mean MS-SSIM of 0.7642, the NANO-TEM results (Sec. A.7 and A.9) reflect the difficulty of reproducing atomic-scale features compared to the larger-scale structural geometries of the DEVICE-TEM dataset. Additionally, our approach requires substantial computational resources, see Sec. A.5, and careful hyperparameter tuning, as the models were developed from scratch.

Future work should focus on integrating advanced guidance mechanisms such as classifier-free guidance (Ho & Salimans, 2022) and model guidance (Tang et al., 2025). While classifier guidance provides controllability for existing pre-trained models, classifier-free and model guidance require integration during the initial model training phase. Both methods eliminate the need for external classifiers, offering improved trade-offs between control and diversity, along with simplified deployment. Furthermore, coupling the generative approach with physical models of electron-specimen interactions could facilitate hybrid approaches that bridge the evident gap between the imaging domain and physically accurate modeling. Investigating the incorporation of diverse noise characteristics (such as Poisson-Gaussian, shot, spatially correlated, structured, contrast and signal-dependent noise, etc.) may broaden the applicability of this approach and open new directions for future research. The integration of synthetic image generation with automated acquisition and analysis workflows represents a natural progression toward semi- or fully-autonomous characterization systems that capitalize on enhanced data availability (Spurgeon et al., 2021; Meirovitch et al., 2023). When combined with real-time acquisition guidance and intelligent sampling strategies, this integration could optimize the trade-off between imaging time and information content in production environments. Additionally, foundation models and their fine-tuning have emerged as promising areas for further investigation.

## 7 CONCLUSION

This work demonstrates a significant advancement over conventional generative methods by successfully generating high-fidelity, high-resolution synthetic TEM images of size $1024 \times 1024$ pixels from extremely limited datasets. The proposed progressive patch-based generative framework has been experimentally validated using two real wafer TEM datasets. Synthetic images generated by this framework achieved higher mean MS-SSIM scores, demonstrating strong preservation of the essential structural and statistical properties inherent to both TEM datasets, properties that are critical for reliable semiconductor metrology. The enhanced data availability achieved through synthetic image generation, while preserving the complex variations inherent in experimental TEM imaging, aims to improve the training of robust downstream models for defect inspection, semantic segmentation, and metrology applications. These applications are typically constrained by the destructive and costly nature of TEM data acquisition, which limits the availability of diverse datasets. By directly addressing this data scarcity, our work enables broader adoption of machine learning-driven methods in advanced semiconductor manufacturing metrology.

ACKNOWLEDGMENT

The authors acknowledge the use of the DDPM implementation by Phil Wang (github.com/lucidrains/denoising-diffusion-pytorch/) as the foundation of their diffusion model architecture. Code prototyping was assisted by Microsoft Copilot 365, using models based on OpenAI GPT-4 and GPT-5. All outputs were reviewed and validated by the authors.

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

# A APPENDIX

## A.1 DIFFUSION FRAMEWORK

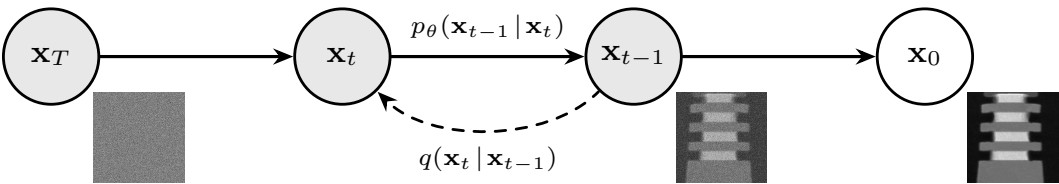

Figure 8: Framework of the forward and reverse diffusion process, adapted from Ho et al. (2020).

## A.2 MODEL ARCHITECTURE

Table 4: U-Net architecture for progressive training

| Stage | Resolution | Channels | ResNet Blocks | Attention Type | Flash Attention |
|-------|-----------|----------|---------------|----------------|-----------------|
| 0 | $1024 \times 1024$ | 64 | 2 | Linear | – |
| 1 | $512 \times 512$ | 64 | 2 | Linear | – |
| 2 | $256 \times 256$ | 128 | 2 | Linear | – |
| 3 | $128 \times 128$ | 256 | 2 | Linear | – |
| 4 | $64 \times 64$ | 512 | 2 | Linear | – |
| 5 | $32 \times 32$ | 512 | 2 | Linear | – |
| 6 | $16 \times 16$ | 512 | 2 | Linear | – |
| 7 | $16 \times 16$ | 512 | 2 | Full | Training only |
| Mid | $16 \times 16$ | 512 | 2 | Full | Training only |

## A.3 DATA (NANO-TEM)

**NANO-TEM Dataset:** 539 high-magnification (960,000×) TEM images with a resolution of ($2048 \times 2048$) pixels, primarily focused on nanosheet and multilayer stacks. These images capture fine structural details at atomic or near-atomic resolution, enabling detailed analysis of layer thickness, interlayer spacing, and crystalline quality. The field of view is limited to a single structure. The dataset features three different imaging modes (BF, ADF, HAADF) across 180 structures, with one missing observation. Fig. 1 (d-f) provides an overview of the modes featured in this dataset.

## A.4 Augmentation Algorithm

---

**Algorithm 1** TrivialAugment procedure for TEM images

---

1: **procedure** TA($x$: TEM image)
2:     Sample an augmentation $a$ from $\mathcal{A}$
3:     Sample a strength $m$ from $\{0, \ldots, 30\}$
4:     Apply rejection criteria for extreme values
5:     Reject if duplicate
6:     **return** $a(x, m)$
7: **end procedure**

---

## A.5 Computational Requirements

### A.5.1 Training

We experienced a minimum of 40 GB VRAM required for batch-size 1, full-scale ($1024 \times 1024$) training. Our implementation utilized $2\times$ NVIDIA 80 GB A100 GPUs for the final training stage with batch-size 2, with batch-sizes increased by a factor of four at each smaller patch resolution stage. The model trained on the smaller DEVICE-TEM dataset converged after 117,500 iterations at 8.96 seconds per iteration, resulting in a total training time of 292.45 hours for the final ($1024 \times 1024$) stage. After training with an initial learning rate of $3 \times 10^{-5}$, the last 2,500 iterations comprised of a fine-tuning phase at a reduced learning rate of $3 \times 10^{-6}$, for which we reduced the occurrence of samples in the training set that were over-represented during sampling. With the larger NANO-TEM dataset, the model converged after 225,000 iterations at 15.2 seconds per iteration, resulting in a total training time of 950 hours. For the ($64 \times 64$) to ($512 \times 512$) stages, training was limited to 10 dataset passes, corresponding to less than 24 hours and approximately 48 hours of additional training time, respectively. Due to computational constraints, we did not employ evaluation metrics such as Fréchet Inception Distance (FID) or Inception Score (IS) for convergence assessment. Instead, we relied on periodic visual inspection of generated samples and subsequent quantitative analysis using the PSNR as detailed in Sec. 4.5. For the larger NANO-TEM model, training could have extended further. However, both visual evaluation and similarity metrics indicated that the generated samples achieved sufficient quality for our application requirements.

### A.5.2 Inference

We observed an inference duration of 0.16 seconds per DDIM timestep (Nichol & Dhariwal, 2021) on an NVIDIA A100 40 GB GPU at a batch-size of 1, while a batch-size of up to 6 was achievable.

## A.6 Structural Similarity Metrics - Formulas

The Peak Signal-To-Noise Ratio (PSNR) for the original images $f$ and the synthetic images $g$ is defined as:

$$\text{PSNR}(f, g) = 10 \times \log_{10}\left(\frac{(2^n - 1)^2}{\text{MSE}(f, g)}\right) \tag{6}$$

where $n$ denotes the maximum value at a given bit-depth and MSE is defined as:

$$\text{MSE}(f, g) = \frac{1}{M \times N} \sum_{i=1}^{M} \sum_{j=1}^{N} (f_{ij} - g_{ij})^2 \tag{7}$$

where $M \times N$ represents the image. A higher PSNR-value indicates similarity to the original image. The Structural Similarity Index Measure (SSIM) (Wang et al., 2004), is defined as:

$$\text{SSIM}(f, g) = l(f, g) \cdot c(f, g) \cdot s(f, g) \tag{8}$$

The SSIM compares images regarding their luminance (l), contrast (c) and structure (s), defined as:

$$l(f,g) = \frac{2\mu_f\mu_g + C_1}{\mu_f^2 + \mu_g^2 + C_1}$$

$$c(f,g) = \frac{2\sigma_f\sigma_g + C_2}{\sigma_f^2 + \sigma_g^2 + C_2} \qquad (9)$$

$$s(f,g) = \frac{\sigma_{fg} + C_3}{\sigma_f\sigma_g + C_3}$$

The individual functions 9 are calculated using the mean luminance $\mu_f$ and $\mu_g$, standard deviation $\sigma_f$ and $\sigma_g$ and covariance $\sigma_{fg}$ of $f$ and $g$. $C_{1-3}$ are constants stabilizing the denominators. A SSIM of 1 indicates identity, 0 indicates no structural correlation.

The MS-SSIM (Wang et al., 2003) improves robustness by computing the SSIM across progressively down-sampled and low-pass-filtered scales and is defined as:

$$\text{MS-SSIM}(f,g) = [l_m(f,g)]^{\alpha_M} \cdot \prod_{j=1}^{M} [c_j(f,g)]^{\beta_j} [s_j(f,g)]^{\gamma_j} \qquad (10)$$

where the exponents $\alpha_M$, $\beta_j$, and $\gamma_j$ weight the relative importance of each component and scale and are practically set equal within each scale to simplify parameter selection. Luminance $l(f,g)$ is calculated only at the coarsest scale M, as comparison is more meaningful at lower resolutions where local pixel variations are smoothed out.

The LPIPS metric (Zhang et al., 2018) leverages features from multiple layers of pre-trained networks (VGG, AlexNet, SqueezeNet) to measure perceptual similarity. It computes weighted L2 distances between channel-wise normalized features:

$$\text{LPIPS}(f,g) = \sum_l \frac{1}{H_l W_l} \sum_{h,w} ||\mathbf{w}_l \odot (\hat{\mathbf{y}}_{f,hw}^l - \hat{\mathbf{y}}_{g,hw}^l)||_2^2 \qquad (11)$$

where $\mathbf{w}_l$ represents learned linear weights for layer $l$, calibrated on human perceptual judgments, $\hat{\mathbf{y}}^l$ indicates channel-wise unit normalized feature maps with spatial dimensions $(H_l, W_l)$, and $\odot$ denotes the Hadamard product. Smaller LPIPS values indicate higher perceptual similarity.

## A.7 Structural Similarity (NANO-TEM)

As Tab. 5 summarizes, our analysis shows moderate performance. We achieve a mean MS-SSIM of 0.7642, indicating reasonable similarity. SSIM shows moderate performance with a mean of 0.4800, while PSNR results in a mean of 22.32 dB. The MSE of 0.0061 is higher than optimal, as is the LPIPS of 0.2420, reflecting the less converged state of the model. We attribute this to the larger dataset size requiring more extensive training iterations. As noted in our computational analysis A.5, additional training of the NANO-TEM model could have led to better convergence. The moderate performance also underscores the challenge of reproducing atomic-scale features compared to structural geometries present in the DEVICE-TEM dataset. Still, the generated images maintain sufficient quality for downstream applications (such as denoising, segmentation, and defect inspection etc.), evidenced by visual assessment.

Table 5: Structural similarity metrics for synthetic NANO-TEM PSNR-sorted subset ($n = 539$)

| Metric | Mean | Std | Min | Max |
|---|---|---|---|---|
| MS-SSIM | 0.7642 | 0.0313 | 0.7187 | 0.8772 |
| SSIM | 0.4800 | 0.0624 | 0.3558 | 0.6667 |
| PSNR (dB) | 22.3173 | 1.2366 | 20.0237 | 28.5515 |
| MSE | 0.0061 | 0.0016 | 0.0014 | 0.0099 |
| LPIPS | 0.2420 | 0.0386 | 0.1131 | 0.3017 |

## A.8   Noise Estimation Metrics - Formulas

We estimated noise standard deviation using the Laplacian operator method (Immerkær, 1996), where an image $f$ with dimensions $H \times W$ and pixel intensities $f_{ij}$, normalized to $[0, 1]$, is convolved with the discrete Laplacian kernel:

$$L = \begin{bmatrix} 0 & -1 & 0 \\ -1 & 4 & -1 \\ 0 & -1 & 0 \end{bmatrix} \tag{12}$$

and the result is denoted as $y = f * L$. The noise standard deviation is then estimated as:

$$\sigma_n(f) = \sqrt{c \cdot \frac{1}{HW} \sum_{i=1}^{H} \sum_{j=1}^{W} y_{ij}^2} \tag{13}$$

where $c$ describes a correction factor accounting for the discrete nature of the Laplacian operator. Higher values of $\sigma_n$ indicate more noise.

The Signal-to-Noise Ratio (SNR) quantifies the ratio of signal strength and noise. It is computed as:

$$\text{SNR}_{\text{dB}}(f) = 20 \log_{10} \left( \frac{\mu_f}{\sigma_n(f)} \right) \tag{14}$$

where $\mu_f$ is the mean intensity of the image. Higher SNR values indicate better signal quality in relation to noise levels.

We define the ratio of High-Frequency Noise (HFNR) as the proportion of high-frequency content in an image, typically corresponding to noise, and compare the average magnitude of high-frequency components to the overall average magnitude:

$$\text{HFNR}(f) = \frac{\text{mean}(|F_{uv}| \text{ high frequencies})}{\text{mean}(|F_{uv}| \text{ all frequencies})} \tag{15}$$

Here, $F$ represents a 2D discrete Fourier transform of $f$ and $|F_{uv}|$ denotes the magnitude of the Fourier coefficient at frequency coordinates $(u, v)$. High frequencies are defined as those outside a centered disk of radius $\lfloor \min(H, W)/4 \rfloor$. Values greater than 1 indicate proportionally elevated high-frequency content, suggesting increased noise.

## A.9   Noise Metrics (NANO-TEM)

Tab. 6 presents our results. The synthetic images exhibit a significantly lower noise standard deviation ($0.016 \pm 0.004$ vs. $0.041 \pm 0.016$, $p < 0.001$) and substantially higher SNR ($27.02 \pm 3.43$ dB vs. $20.37 \pm 3.18$ dB, $p < 0.001$), corresponding to an improvement of approximately 6.65 dB. The high-frequency noise ratio shows only minimal variation ($1.128 \pm 0.013$ vs $1.121 \pm 0.027$, $p < 0.001$).

The noise reduction likely reflects the model's challenge in accurately replicating the complex noise environment of atomic-resolution imaging. While the SNR improvement is potentially beneficial for certain applications, we clearly observe how the additional dataset passes contributed to the DEVICE-TEM model's enhanced overall alignment with the domain.

Table 6: Noise metrics for synthetic NANO-TEM PSNR-sorted subset ($n = 539$)

| Metric | Original Mean $\pm$ SD | Synthetic Mean $\pm$ SD | p-value |
|--------|--------|--------|--------|
| Noise std | $0.041 \pm 0.016$ | $0.016 \pm 0.004$ | $< 0.001$ |
| SNR (dB) | $20.37 \pm 3.18$ | $27.02 \pm 3.43$ | $< 0.001$ |
| HFNR | $1.121 \pm 0.027$ | $1.128 \pm 0.013$ | $< 0.001$ |

## A.10   Patch-based Training and Layer Freezing Ablation Study

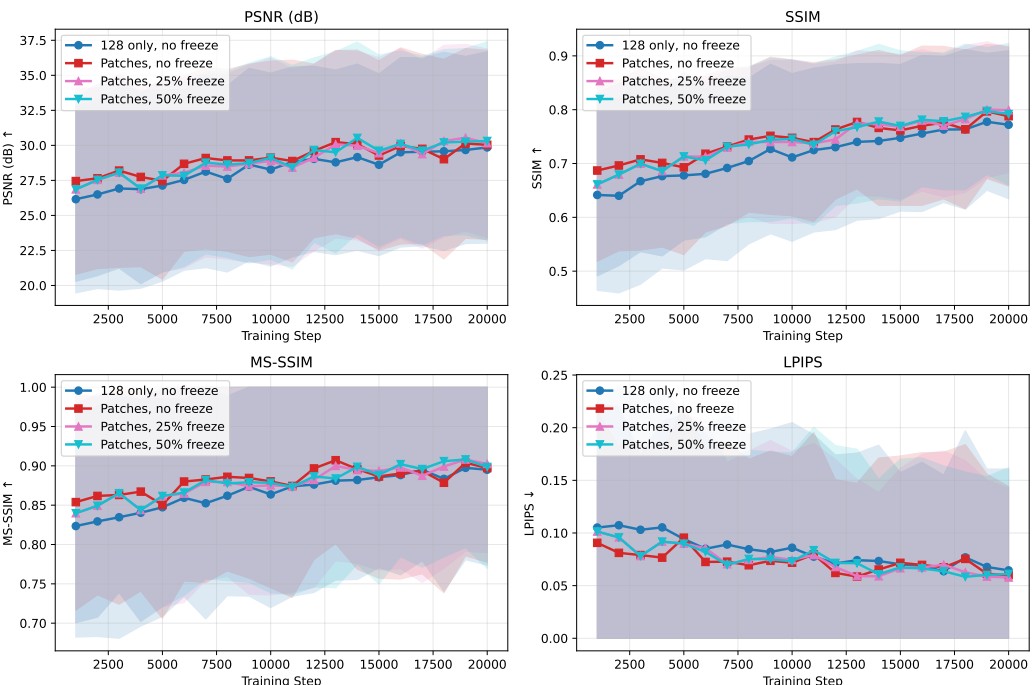

Figure 9: Ablation study evaluating the proposed patch-based DDPM training and the layer-freezing strategy against unmodified baseline DDPM training. The patch-based models were initially trained for 10,000 training steps at a $64 \times 64$ resolution using the DEVICE-TEM dataset. The $128 \times 128$ resolution stage lasted 20,000 training steps. Each 1,000 trainings steps of the $128 \times 128$ resolution stage, evaluation metrics were computed on 288 samples and are reported as mean $\pm$ std. Our analyses indicate that freezing 25% or 50% of the network layers leads to improved performance over the baseline DDPM and over patch-based training at later iterations (15k–20k), with freeze-50 providing the strongest effect (though still relatively modest).

## A.11 CLASSIFIER GUIDANCE UMAP

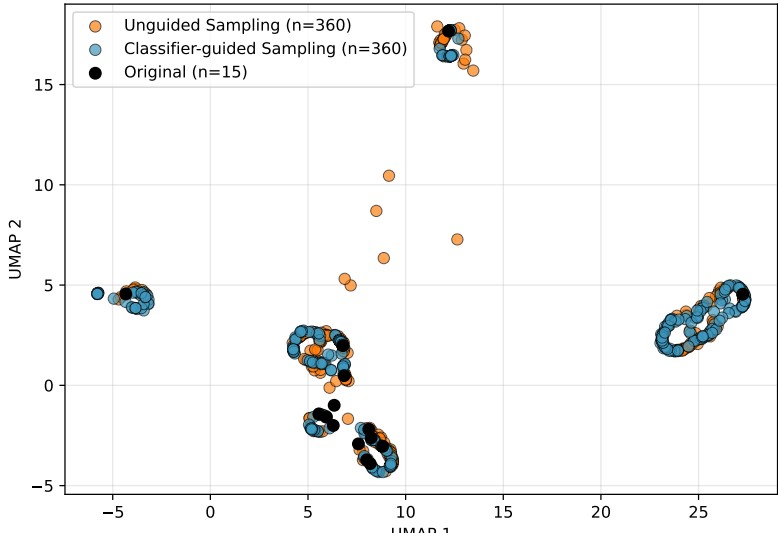

Figure 10: UMAP of DEVICE-TEM samples generated without guidance and with guidance scale ($s \in [0.5, 10]$, $\Delta s = 0.5$) with 15 samples for each scale.

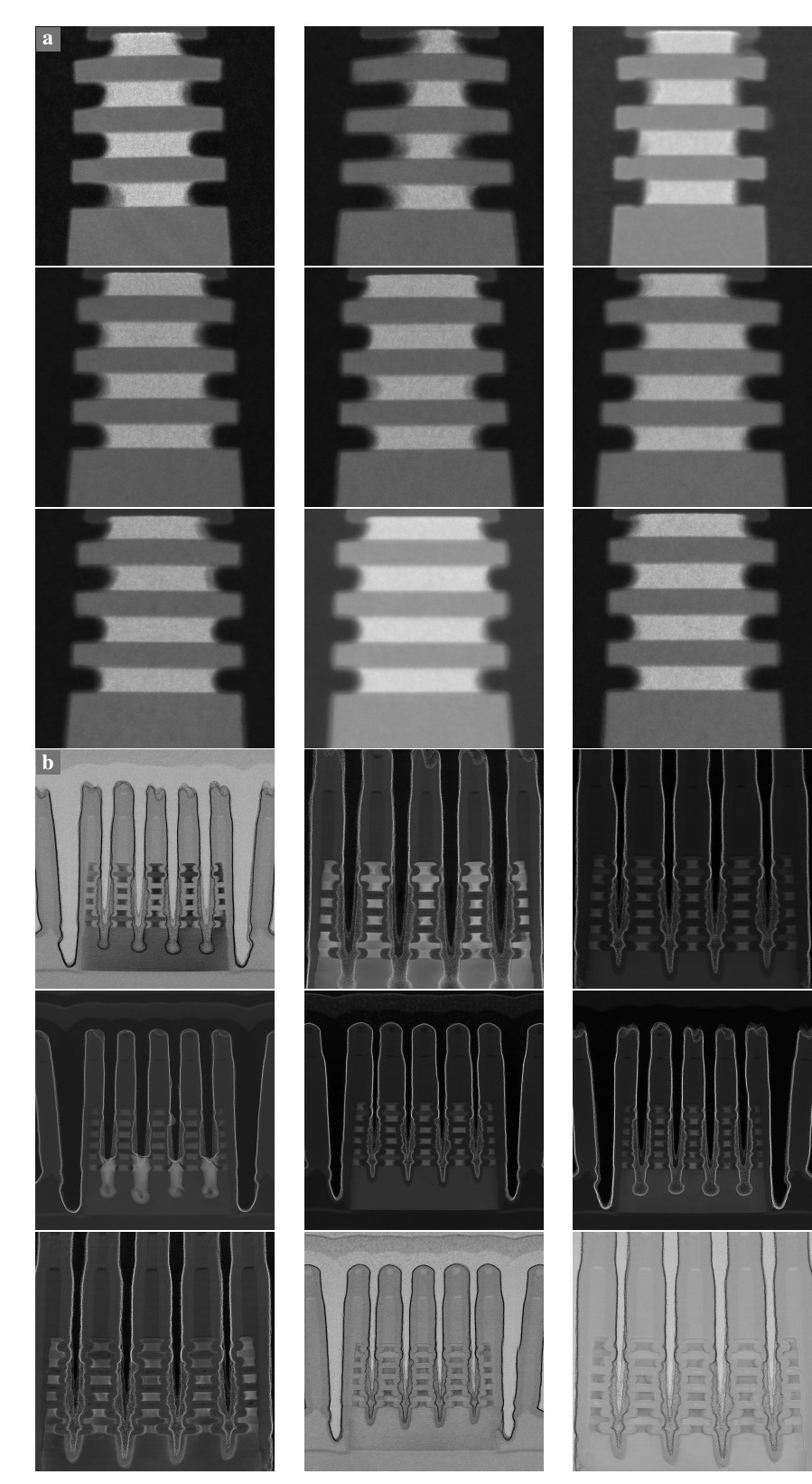

Figure 11: Original (a, b) and synthetic NANO-TEM and DEVICE-TEM images.

