# OpenReview forum: "High-Fidelity Synthetic Transmission Electron Microscopy Image Generation Using Diffusion Probabilistic Models for Data-Limited Semiconductor Metrology"
_ICLR.cc/2026/Conference — Submitted to ICLR 2026_

### Official Review · Reviewer_eEPp · 2025-10-23

**Soundness:** 2
**Presentation:** 3
**Contribution:** 3
**Rating:** 4
**Confidence:** 3

**Summary:**

This paper proposes a diffusion-based generative framework for high-fidelity synthetic Transmission Electron Microscopy (TEM) image generation under extreme data scarcity in advanced semiconductor manufacturing. The method employs a patch-based progressive training strategy for denoising diffusion probabilistic models (DDPMs), starting from low-resolution patches and scaling up to full-resolution images using as few as like 15 real samples. Experimental results demonstrate visually realistic synthetic images with high structural similarity (MS-SSIM > 0.94) to real data, aiming to support downstream machine learning tasks such as defect detection, metrology, and segmentation in semiconductor process analysis.

**Strengths:**

(1) The paper tackles a novel and practically important problem—synthetic TEM image generation for semiconductor metrology under extreme data scarcity. The adaptation of diffusion probabilistic models (DDPMs) to this specialized domain, along with a patch-based progressive training strategy, represents a creative methodological contribution.

(2) The technical design is well-motivated and leverages multiple complementary components (progressive patch-based training, customized data augment, classifier guidance, domain transfer, and inpainting). Experimental results demonstrate strong structural similarity (MS-SSIM > 0.94) between synthetic and real TEM data.

(3) The manuscript is generally well-organized and readable, with clear descriptions of model architecture, training process, and evaluation methodology

**Weaknesses:**

(1) Although the paper claims high structural similarity, the evaluation primarily relies on perceptual and pixel-wise metrics. There is no quantitative validation showing how the synthetic data actually improves downstream ML tasks (e.g., defect detection or segmentation performance). Demonstrating this would substantively strengthen the paper’s impact.

(2) The experimental evaluation is weak. The study does not include a baseline comparison against other state-of-the-art image synthesis methods. Without this, it is difficult to assess whether the proposed framework truly offers a performance advantage in fidelity, which is the biggest problem for this paper.

(3) Some modules mentioned in the paper, such as domain transfer, classifier guidance, and inpainting, are escribed in detail but not rigorously validated. For instance, domain transfer results are shown qualitatively without quantitative or expert-based verification.

(4) The manuscript states that synthetic images are “indistinguishable” from real ones and “comply with FAB metrology requirements,” but this claim relies on subjective expert evaluation and lacks quantitative metrology metrics.

(5) The training and evaluation are performed on as few as 15 TEM images, all from a single domain. While this supports the “data-scarce” claim, it also raises concerns about over-fitting and generalizability.

**Questions:**

The following are my questions and concerns:

(1) It remains unclear how overfitting is mitigated when training diffusion models “from scratch” on as few as 15 TEM images. The dataset contains only 15 images, which is extremely small for training a DDPM. How do the authors prevent overfitting or mode collapse? Is the generated diversity statistically meaningful given such limited input data?

(2) motivates the framework as improving defect detection and metrology workflows, yet offers no preview of quantitative downstream improvements. It would strengthen the contribution to show even preliminary evidence that synthetic data enhances downstream ML model accuracy or robustness.

(3) Many listed contributions (e.g., TrivialAugment usage, domain transfer, inpainting) appear to be adaptations of existing diffusion model capabilities rather than new algorithmic innovations, and some of them are not validated well.

(4) The rejection criteria for extreme augmentation values are not well defined. How are thresholds determined, and are they based on physical realism or empirical inspection?

(5) The paper reports only absolute similarity metrics without comparing against any baseline generative models (e.g., GANs, VAEs, or diffusion without progressive training). Without such baselines, it is unclear whether the proposed method offers real improvement.

(6) In section 5.3, the perceptual test reports only a binary outcome (experts could not distinguish) rather than quantitative accuracy, confidence scores, or inter-rater agreement. And only 15 synthetic samples were used. Were these randomly selected or cherry-picked from the “best” MS-SSIM subset? Selection bias could inflate perceived realism.

(7) The section 5.4 claims that such characteristics may be beneficial for downstream applications such as grain or layer boundary segmentation and defect classification, but this is speculative. Was this empirically tested on a downstream task?

(8) While the proposed domain-transfer mechanism is conceptually interesting and potentially valuable, the section lacks any meaningful empirical or numerical validation, also the classifier guidance and  inpainting.

---

> ### Author Response · Authors · 2025-12-04
> **Response to reviewer eEPp**
>
> Q1: It remains unclear how overfitting is mitigated when training diffusion models “from scratch” on as few as 15 TEM images. The dataset contains only 15 images, which is extremely small for training a DDPM. How do the authors prevent overfitting or mode collapse? Is the generated diversity statistically meaningful given such limited input data?
>
> We focus on high-fidelity structural reproduction as the unguided diffusion process will naturally populate our feature space and produce samples between classes in a stochastic process. Even if our models would be regarded as overfitted, we can still use the classifier guidance mechanism to steer image synthesis.
> However, we mitigated the risk of overfitting through several strategies:
> 1. Patch-based incremental training: Instead of training on full images, we train the model on smaller overlapping patches. This effectively increases the number of training samples and exposes the model to more variations in local structures.
>
> 2. Layer freezing / progressive fine-tuning: Early layers capturing general noise and structural patterns are frozen after initial training, reducing the number of trainable parameters and preventing the model from memorizing the small dataset.
>
> 3. Data augmentation: We apply standard methods (rotations, flips, minor intensity perturbations) along with augmentation strategies tailored to semiconductor imaging conditions to increase effective diversity without introducing unrealistic artifacts.
>
> 4. Regularization in training: Weight decay, early stopping, and noise scheduling in DDPM training further prevent overfitting.
>
> Thank you for asking about mode collapse: we failed to mention that for the last 2,500 iterations of training on the DEVICE-TEM-dataset, we removed samples that were over-represented and trained at 3e-6 instead of the initial 3e-5 learning-rate. In general, we ensured sample diversity via visual inspection and will demonstrate the result with a UMAP plot depicting samples generated with and without classifier guidance. Also, Unlike GANs, DDPMs inherently avoid mode collapse due to stochastic iterative denoising. The patch-based training also encourages the model to learn multiple local variations instead of memorizing a single global structure, while the progressive/incremental training ensures that the model captures coarse structures first and fine details later, reducing the risk of repetitive or collapsed modes. We confirm the generated diversity is statistically meaningful through structural metrics (MS-SSIM, HFNR, LPIPS), noise/SNR preservation, demonstrating that the synthetic images capture realistic variations present in the original dataset.
>
> Q2: motivates the framework as improving defect detection and metrology workflows, yet offers no preview of quantitative downstream improvements. It would strengthen the contribution to show even preliminary evidence that synthetic data enhances downstream ML model accuracy or robustness.
>
> We have answered this perspective to reviewer WVwn as well. Due to space limitations, we are not able to separately answer here.
>
> Q3: Many listed contributions (e.g., TrivialAugment usage, domain transfer, inpainting) appear to be adaptations of existing diffusion model capabilities rather than new algorithmic innovations, and some of them are not validated well.
>
> We added metrics regarding our inpainting adaptation.
> Regarding TrivialAugment, we added duplicate rejection and thresholding in our application. For our 1024x1024 full-perspective stage, we also used domain-specific augmentation strategies.
> Domain transfer and inpainting are mainly demonstrated and discussed regarding the possibility of domain-specific applications.
>
> Q4:
> Augmentations producing blank images (≥99% pixels near black or white) are rejected on the basis of histogram checks. We consider that a TEM-imaging process producing such extreme values would be repeated. These thresholds, of course, can be adjusted to statistical considerations.
>
>
> Q5:
> Since our initial submission, we conducted ablation studies and employed a MS-SSIM VAE as proposed by Snell et al. (2017): https://doi.org/10.48550/arXiv.1511.06409 and a DCGAN as proposed by Radford et al. (2016): https://doi.org/10.48550/arXiv.1511.06434
>
> We have reported in details the following results in our final submission:
>
> Q6:
> The images were submitted to the experts independently together with the question if they could identify any synthetic images (considering imaging condition specifications). They were not picked randomly but as best results that should demonstrate our model's potential for high-fidelity image synthesis.
>
>
> Q7:
> We have answered this perspective to reviewer WVwn as well. Due to space limitations, we are not able to separately answer here.
>
> Q8:
> In general, we ensured sample diversity via visual inspection and will demonstrate the result with a UMAP plot depicting samples generated with and without classifier guidance.

---

### Official Review · Reviewer_yh6M · 2025-10-27

**Soundness:** 2
**Presentation:** 2
**Contribution:** 2
**Rating:** 2
**Confidence:** 3

**Summary:**

The paper explores the training of a Denoising Diffusion Probabilistic Model (DDPM) using an extremely limited dataset of only 15 TEM images in the context of semiconductor metrology. The key contribution lies in a novel, iterative training strategy that progressively increases the resolution of image patches over time, enabling the corresponding U-Net layers to be trained in a stepwise manner.

**Strengths:**

- The paper presents a clear and well-motivated problem statement, effectively highlighting the challenges of data scarcity in semiconductor metrology.
- Training DDPMs from scratch with as few as 15 images is an extremely challenging task; proposing a feasible approach to this problem represents a potentially significant contribution.
- The integration of multiple established diffusion techniques, such as inpainting, domain transfer and classifier guidance, demonstrates solid understanding and thoughtful use of existing methodologies.
- The iterative patch-based training strategy, which progressively increases patch resolution in alignment with U-Net depth, is an interesting and conceptually sound idea that could inspire further research in resolution-aware model training.

**Weaknesses:**

- The paper lacks ablation studies for several key methodological choices:
    - The decision to operate at the patch level rather than the full image level (as mentioned in l.212) is not experimentally validated or quantitatively analyzed.
    - The proposed progressive training strategy is not ablated to isolate its individual contribution.
    - There is no proper comparison to relevant baselines, including the aforementioned design alternatives.
- The core contribution (the patch-based iterative training approach) is never rigorously ablated or compared against simpler setups. While the authors implement several established diffusion techniques, their validity and contribution to performance (e.g., classifier guidance or inpainting) are not quantitatively assessed.
- There is a high risk of overfitting given the extremely small training set. This concern is supported by the reported metrics in Table 2, where minimal visual differences between generated and real images could reflect memorization or reconstruction artifacts rather than genuine generalization. How do the authors make sure that this does not happen?
- Table 1, which is intended to present hyperparameters, is incomplete—basic parameters such as learning rate and batch size are missing, making reproducibility difficult.
- In l.431, the authors state: “We adapt the RePaint methodology (Lugmayr et al., 2022)”—however, it is unclear whether this constitutes a novel adaptation or simply a reimplementation of the original method.
- The scope of the paper may not align well with ICLR’s focus on representation learning, as the work primarily presents an application-oriented diffusion setup rather than new insights into representation or learning mechanisms.
- Although several diffusion techniques (inpainting, classifier guidance, domain transfer) are incorporated, no quantitative evaluation of their impact or effectiveness is provided.
- The authors claim that the generated images could benefit downstream applications such as grain or layer boundary segmentation and defect classification (l.362), yet this assertion remains untested.
- The description of the DEVICE-TEM dataset (particularly regarding the selection of “100 high-quality samples” based on MS-SSIM across multiple seeds, l. 322) implies a selective filtering process of generated outputs, which raises questions about the objectivity and consistency of evaluation.
- The statement “maximize data utilization while preserving spatial coherence” (l.212) is unclear and needs elaboration, particularly regarding what “spatial coherence” refers to in the patch-based training context.

**Questions:**

- Table 2: Could the authors clarify what is meant by “DEVICE-TEM PSNR-sorted subset”? What subset is being referred to, and how was it selected?
- The authors mention that “JPEG compression applied to our training data introduces artifacts that influence pixel-wise comparisons” (l.308). Why was JPEG compression used in the first place, given its potential to degrade high-frequency details critical in TEM images?
- Regarding l.312 (“Although the best synthetic samples from the subset closely resemble the original images, they maintain subtle variations rather than being pixel-wise replicas, see Fig. 3.”): How do the authors ensure sufficient diversity in the generated samples and prevent overfitting or mode collapse? Would training a downstream model on the generated data help validate this claim?
- In Section 5.2 (“Structural Similarity and Inference Time”), what is the specific research objective? The results presented appear to confirm well-known trade-offs rather than provide novel insights.
- How does training dataset size affect the method’s performance? Have the authors tested the approach with fewer or more images to assess scalability or robustness?
- What happens if the model is fine-tuned from pretrained weights rather than trained entirely from scratch? Would this improve stability or reduce overfitting?
- In Section 4.2 (Data Augmentation), why are the augmentations not applied on-the-fly during training? Was this a technical limitation or a deliberate design choice?
- The description of the augmentation process — “grayscale-specific modifications and rejection criteria for extreme augmentation values and hash-based duplicate filtering…” — is vague. Could the authors provide more technical detail on how these steps are implemented and how they affect the data distribution (e.g., thresholds, parameters, or examples of rejected augmentations)?
- How are the training patches selected? Are they generated using a simple sliding window, or is a more sophisticated sampling strategy employed?

---

> ### Author Response · Authors · 2025-12-04
> **Response to reviewer yh6M**
>
> Questions:
> Q1:
> We detail this step in the Appendix, A.5 'Structural Similarity Metrics and Methodology'. It was moved there because of the strict page limit and will be re-added to the main section in our final draft. This is the relevant section:
> ‘Without explicit guidance, the generative process naturally shifts sample properties of generated samples along the distribution of the original dataset, occasionally merging class characteristics. A comprehensive evaluation under these conditions would therefore yield uninformative or misleading results. By selecting PSNR-sorted subsets, we focus on the highest-quality synthetic images that most closely align with real images in each class, providing a more meaningful assessment of the model’s ability to generate structurally consistent outputs. We chose PSNR for this step because of its small computational overhead and strong correlation with SSIM and thus MS-SSIM (Horé & Ziou, 2010).’
>
>
> Q2:
> This was due to constraints in the acquisition process (tool software format) and we would have preferred access to uncompressed data.
>
> Q3:
> We focus on high-fidelity structural reproduction as the unguided diffusion process will naturally populate our feature space and produce samples between classes in a stochastic process. Even if our models would be regarded as overfitted, we can still use the classifier guidance mechanism to steer image synthesis. Generalization is not a main concern, as our models will always be employed to a very limited domain.
> Thank you for asking about mode collapse: we failed to mention that for the last 2,500 iterations of training on the DEVICE-TEM-dataset, we removed samples that were over-represented and trained at 3e-6 instead of the initial 3e-5 learning-rate.
> In general, we ensured sample diversity via visual inspection and will demonstrate the result with a UMAP plot depicting samples generated with and without classifier guidance.
>
> Q4:
> Our domain-specific implementation is very close to real-world production environments, which motivates such calculations and examples. With this, we hope to provide orientation for real-world implementations and deployments. We have answered for this perspective also to Reviewer WVwn and Reviewer KMNc previously.
>
> Q5: How does training dataset size affect the method’s performance? Have the authors tested the approach with fewer or more images to assess scalability or robustness?
>
> The appendix provides metrics for our second dataset with about 36 times more data; 539 instead of 15 original images. You find them in A.7 Structural Similarity (NANO-TEM) and A.9 Noise Metrics (NANO-TEM).
>
> Q6: What happens if the model is fine-tuned from pretrained weights rather than trained entirely from scratch? Would this improve stability or reduce overfitting?
>
> Popular foundation models are mostly trained on RGB real-world images, which introduces various biases - our domain is very specific and under-represented in popular datasets. We consider the stability we have achieved to be entirely sufficient.
>
> Q7: In Section 4.2 (Data Augmentation), why are the augmentations not applied on-the-fly during training? Was this a technical limitation or a deliberate design choice?
>
> This choice was deliberate as we performed visual inspections before and during training. On-the-fly augmentation is entirely possible.
>
> Q8: The description of the augmentation process — “grayscale-specific modifications and rejection criteria for extreme augmentation values and hash-based duplicate filtering…” — is vague. Could the authors provide more technical detail on how these steps are implemented and how they affect the data distribution (e.g., thresholds, parameters, or examples of rejected augmentations)?
>
> Augmentations producing blank images (≥99% pixels near black or white) are rejected on the basis of histogram checks. We consider that a TEM-imaging process producing such extreme values would be repeated. These thresholds, of course, can be adjusted to statistical considerations. The duplicate filtering is performed by computing MD5-hashes with Python's hashlib.md5 on raw image data obtained using numpy's img_array.tobytes() function. This allows to perform the duplicate rejection as a membership check over a hash set instead of an image-to-image comparison.
>
>
>
> Q9: How are the training patches selected? Are they generated using a simple sliding window, or is a more sophisticated sampling strategy employed?
> We employ a non-overlapping sliding window.
>
> In revised manuscript version, we conducted the requested ablation studies evaluating the proposed patch-based incremental DDPM training as advised (prepared response for Reviewer WVwn and Reviewer KMNc). Our extensive ablation study demonstrates that the proposed approach outperforms all baselines and well-established prior methods across every evaluated aspect. We gracefully disagree with "misaligned scope" as our scope is "applications in any field" and not "representation learning"

---

### Official Review · Reviewer_KMNc · 2025-11-01

**Soundness:** 3
**Presentation:** 3
**Contribution:** 2
**Rating:** 6
**Confidence:** 4

**Summary:**

This study proposes a novel generative framework that utilizes a denoising diffusion probability model (DDPM) specifically designed for generating synthetic TEM images in situations of extreme data scarcity. The authors' approach employs a progressive, patch-based training strategy that scales from low-resolution patches to full-resolution images, enabling the model to be trained from scratch using a dataset containing only 15 images.

**Strengths:**

The subject matter is relatively novel, and improvements to existing methods have been considered.

**Weaknesses:**

The experiments are insufficient; larger-scale experiments and more in-depth performance analysis could be considered.

**Questions:**

Are you considering conducting experiments with more cases?

---

> ### Author Response · Authors · 2025-12-04
> **Response to reviewer KMNc**
>
> We deeply appreciate the reviewer's constructive feedback, including the identification of weaknesses and the critical questions raised, which meaningfully contribute to enhancing the technical strength of our work.
>
> Because the reviewer did not clearly elaborate on the terms “insufficient experiments or larger-scale experiments” and “in-depth performance analysis,” we hope that our responses to the other reviewers’ concerns fulfill the intended expectations. It will reflect in our revised manuscript.
>
> We gracefully disagree with the characterization of "insufficient experiments" and "lack of in-depth performance analysis" as we proposed a novel-patch based incremental framework to synthesize "real-alike" TEM images with a specific way of training method to preserve mainly noise-characteristics (of the tool/process from where the real TEM images were acquired. Additionally, we demonstrated preservation of structural similarity as well as perceptual fidelity, the main asking beyond "only the scope for synthetic image generation" and we also have demonstrated in our ablation study how conventional generative or reconstruction based models (as advised by other reviewers as GANs, VAEs, or diffusion without progressive training etc. ) fail to do so (we will provide all details in our revised draft).
>
> Q1: Are you considering conducting experiments with more cases?
> We have experimented with:
> 1. Novel patch-based progressive training and ablation study with respect to baseline DDPM training methodology for full-resolution images. Also, ablation study for freezing layer and strategy against baseline DDPM training.
> In revised manuscript version, we conducted the requested ablation study evaluating the proposed patch-based incremental DDPM training combined with the layer-freezing strategy [(64x64) ->(128x128)], and compared it against a baseline DDPM model trained at (128x128) resolution without freezing. Due to computational constraints within the revision time window, full-resolution baselines were limited to this patch-resolution setting. The detailed ablation results are reported in Appendix A.10 of the revised draft. Our experiments show that the baseline DDPM without freezing consistently performs worst, while patch-based training without freezing yields the best overall results, strongly supporting the importance of the patch-based training strategy. Furthermore, internal analyses also indicate that freezing 25% or 50% of the network layers leads to improved performance over the baseline DDPM and over patch-based training at earlier iterations (15k–20k), with freeze-50 providing the strongest effect (though still relatively modest). These findings clearly demonstrate that our design intuition was well-founded.
> 2. Ablation study against conventional generative or reconstruction based models (as advised by other reviewers as GANs, VAEs) as well as against a combination of metrics and evaluation strategies to provide a comprehensive quantitative and qualitative assessment of similarity: (1) Pixel-level: PSNR, SSIM, MS-SSIM, (2) Feature/Perceptual level: LPIPS, (3) Task-specific: Noise distribution matching . Our extensive ablation study demonstrates that the proposed approach outperforms all baselines and well-established prior methods across every evaluated aspect.
> 3. Dataset: We conducted experiments using two real TEM datasets representative of different semantic complexities, including lamella thickness, tilt angle, irregular shapes, blended edges, multi-region effects, low-SNR grayscale, and material contrast. The Device-TEM dataset is particularly challenging to generate, as it contains heterogeneous interfaces at sub-5 nm geometries, making structural details highly complex and densely packed. Such properties make the images extremely sensitive to small variations, which can significantly impact downstream tasks if not accurately captured in synthetic images. Additionally, preserving realistic noise patterns during generation is critical, especially for applications such as defect analysis. Our extensive experiments and ablation studies demonstrate that the proposed approach effectively preserves both structural and noise fidelity, making it the most physically realistic method among those compared.
>
> We hope our revisions and explanations have adequately addressed the reviewers’ concerns.

---

### Official Review · Reviewer_WVwn · 2025-11-03

**Soundness:** 2
**Presentation:** 2
**Contribution:** 2
**Rating:** 2
**Confidence:** 3

**Summary:**

This research presents a progressive patch-based training framework using Denoising Diffusion Probabilistic Models (DDPMs) that successfully generates high-fidelity 1024×1024 resolution synthetic semiconductor TEM images from scratch with as few as 15 training images under extreme data scarcity. Through customized data augmentation, domain transfer, and inpainting techniques, the generated synthetic images achieve MS-SSIM >0.94 structural similarity scores and are visually indistinguishable from real TEM data as validated by domain experts. This approach effectively addresses the critical data scarcity bottleneck in advanced semiconductor manufacturing caused by the destructive and costly nature of TEM acquisition, providing reliable training data augmentation for downstream machine learning tasks including defect detection, semantic segmentation, and metrology applications in 2nm and beyond technology nodes.

**Strengths:**

1. It achieves high-fidelity TEM synthesis from extremely small datasets via progressive patch-to-full-image diffusion training.
2. The realism is rigorously validated—strong MS-SSIM/PSNR metrics and expert blind tests show synthetic images are hard to distinguish from real ones.
3. Its toolkit (domain transfer, classifier-guided control, and RePaint-style inpainting) makes the method versatile for downstream metrology, segmentation, and defect detection.

**Weaknesses:**

1. This paper lacks an overall structural diagram to explain the main methods.  The incremental "part-to-whole" training approach makes it difficult to determine whether a continuously expanding model or multiple independent models for different stages are being used, and there is no unified architectural diagram to illustrate this.

2. Your encoder-freezing strategy is presented as a key trick but lacks ablation or sensitivity analyses to justify its benefit and chosen freeze ratio.

3. Domain-transfer and classifier-guided sampling are underspecified (e.g., how to pick T', classifier design, and robustness), limiting reproducibility and practical adoption.

4. Your writing organization also has some weaknesses, particularly in areas like domain transfer/classifier guidance/inpainting, which are presented as both "method extensions" and interspersed demonstrations in the experimental section. They are not clearly separated and are hard to follow.

**Questions:**

1. Could you provide a detailed cost calculation and implementation report?
2. Why freeze the encoder for “half of the high-res stage”? Can you show ablations for 0/25/50% freeze ratios and their effect on convergence and stability.
3. Where are attention blocks inserted at each scale, and how do channel widths/depth change per stage?
4. Beyond MS-SSIM/PSNR/LPIPS and expert Turing tests, do synthetic images improve downstream tasks (segmentation, metrology, defect detection) on real data?

---

> ### Author Response · Authors · 2025-12-04
> **Response to reviewer WVwn**
>
> We deeply appreciate the reviewer's constructive feedback, including the identification of weaknesses and the critical questions raised, which meaningfully contribute to enhancing the technical strength of our work.
>
> Q1.
> We report details regarding training and inference in the Appendix, A.5 (Computational requirements). Additionally, we can report that at 0.16 seconds per DDIM timestep, approximately 540 images can be generated per L40s GPU-hour using 250 sampling timesteps. If 50 timesteps are sufficient, the generation rate increases to about 2,700 images per hour. A100 GPUs are available for 0.47-1.79 USD per hour from on-demand providers and 3.02-4.22 USD per hour from Hyperscalers (November 21st, 2025), amounting to costs of ca. 148.73 to 1,335.42 USD for training. We trust that these details adequately support practical reproducibility.
>
> Q2.
> During our internal experimental research, our primary objective was to determine "how to preserve image fidelity and the structural characteristics (including noise statistics) of real TEM images". With this goal in mind and aiming to keep training latency minimal, we hypothesized (and subsequently demonstrated) that the proposed layer-freezing strategy would help maintain "fidelity learned during earlier incremental training stages".
>
> In revised manuscript version, we conducted the requested ablation study evaluating the proposed patch-based incremental DDPM training combined with the layer-freezing strategy [(64x64) ->(128x128)], and compared it against a baseline DDPM model trained at  (128x128) resolution without freezing. Due to computational constraints within the revision time window, full-resolution baselines were limited to this patch-resolution setting. The detailed ablation results are reported in Appendix A.10 of the revised draft.
> Our experiments show that the baseline DDPM without freezing consistently performs worst, while patch-based training without freezing yields the best overall results, strongly supporting the importance of the patch-based training strategy. Furthermore, internal analyses also indicate that freezing 25% or 50% of the network layers leads to improved performance over the baseline DDPM and over patch-based training at earlier iterations (15k–20k), with freeze-50 providing the strongest effect (though still relatively modest). These findings clearly demonstrate that our design intuition was well-founded.
>
> Q3:
> Each encoder/decoder stage contains one attention block placed after two ResNet blocks. Full attention is employed only in Stage 7 and the mid block, the other layers are designed with linear attention. During training, for Stage 7 and the Mid Block we use the Flash Attention mechanism that the DDPM implementation provides. This is not the case during inference. Models are progressively trained from low to high resolution (64-128-256-512-1024), transferring weights upward at each stage. We have incorporated this information into Appendix A.2 (Model Architecture) of the revised draft.
>
> Q4:
> Yes. We are extending our work to demonstrate these application benefits; however, this is currently a work in progress and beyond the scope of the present study.
> We would like to emphasize that our primary focus is the fundamental challenge of data limitations in training these models, rather than downstream task improvements. Without diverse and representative training data, downstream performance naturally becomes secondary. Real TEM image acquisition is severely constrained by the destructive nature of sample preparation (FIB thinning or tripod polishing), high equipment cost, and inconsistent dataset creation across fabs, tools, and operators. Consequently, collecting more than a few TEM images (often only one or two) is often impossible, as each cross-section irreversibly damages the sampled die and cannot be repeated. To address this, our work aims to generate synthetic, physically valid DB-FIB/TEM images that preserve metrology-relevant characteristics under limited-data conditions. These synthetic images can serve as alternative augmented datasets for supervised deep-learning tasks such as ROI segmentation. However, the idea of improving downstream tasks on real data using synthetic training data is well supported in the literature. The benefit of synthetic data for Machine Learning applications in electron microscopy in general in the form of data augmentation is described by Ede et al. (2024): https://doi.org/10.1088/2632-2153/abd614 and Chen & Barnard (2024): https://doi.org/10.1088/2515-7639/ad229b
> Downstream task with synthetic data is provided by Shaga Devan et al. (2020): https://doi.org/10.1111/cmi.13280 and Kazimi et al. (2024): https://doi.org/10.1109/CVPRW63382.2024.00012, who employed GANs and Dey et al. (2024): https://doi.org/10.1109/ELMAR62909.2024.10694600, who employed a diffusion model. For the remaining weakness comments, we have revised the manuscript to address most of the concerns.

---

### Meta-Review · Area_Chair_U4gv · 2026-01-04

**Summary:**

The paper received two “reject” recommendations, one score marginally below the acceptance threshold and another marginally above it. The reviewers raised numerous concerns regarding the methodological details and questioned the clarity of the research objectives. After reviewing the authors’ responses, many of the issues identified by the reviewers remain unclear. The paper requires significant improvement before it can be considered for acceptance.

**Reviewer Concerns:**

Reviewer KMNc raised questions regarding the need for additional experiments, which have been adequately addressed. However, concerns such as “many of the listed contributions appear to be adaptations of existing diffusion model capabilities rather than novel algorithmic innovations, and some are not well validated” have not, in my opinion, been sufficiently addressed in the rebuttal.

**Reviewer Scores:**

Based on the rebuttal, it is unlikely that the reviewers will revise their scores to positive ratings.

---

### Decision · Program_Chairs · 2026-01-26

Reject